# Beyond Outcome Reward: Decoupling Search and Answering Improves LLM Agents

## Abstract

Enabling large language models (LLMs) to utilize search tools offers a promising path to overcoming fundamental limitations such as knowledge cutoffs and hallucinations. Recent work has explored reinforcement learning (RL) for training search-augmented agents that interleave reasoning and retrieval before answering. These approaches usually rely on outcome-based rewards (*e.g.*, exact match), implicitly assuming that optimizing for final answers will also yield effective intermediate search behaviors. Our analysis challenges this assumption: we uncover multiple systematic deficiencies in search that arise under outcome-only training and ultimately degrade final answer quality, including failure to invoke tools, invalid queries, and redundant searches. To address these shortcomings, we introduce **DeSA** (**De**coupling **S**earch-and-**A**nswering), a simple two-stage training framework that explicitly separates search optimization from answer generation. In Stage 1, agents are trained to improve search effectiveness with retrieval recall-based rewards. In Stage 2, outcome rewards are employed to optimize final answer generation. Across seven QA benchmarks, DeSA-trained agents consistently improve search behaviors, delivering substantially higher search recall and answer accuracy than outcome-only baselines. Notably, DeSA outperforms single-stage training approaches that simultaneously optimize recall and outcome rewards, underscoring the necessity of explicitly decoupling the two objectives.

## 1 Introduction

The inherent factuality limitations of large language models (LLMs), including knowledge cutoffs (Cheng et al., 2024) and hallucinations (Huang et al., 2025), have been increasingly mitigated by leveraging external tools like retrieval systems (Lewis et al., 2020) and web search (Wei et al., 2025c). Early progress came from retrieval-augmented generation (RAG) (Gao et al., 2023), which grounds model outputs in relevant documents through one-shot retrieval. More recent advances push beyond static retrieval toward interactive search agents (Jin et al., 2025; Song et al., 2025) and "Deep Research" agents (Zheng et al., 2025; Sun et al., 2025b) that iteratively reason, issue queries, and refine their outputs through multi-step investigation and execution. Together, these developments mark a shift toward more autonomous, process-aware information seeking.

While early approaches like prompting (Yao et al., 2022) and supervised fine-tuning (SFT) (Song et al., 2024) can elicit such agentic behaviors, they face significant limitations in robustness and scalability: prompting alone is often sensitive to templates and generalizes poorly (Sclar et al., 2024), while SFT depends on costly, human-curated datasets that are difficult to scale (Ouyang et al., 2022). More recent work turns to reinforcement learning (RL) (Kaelbling et al., 1996; DeepSeek-AI et al., 2025; Shao et al., 2024; Schulman et al., 2017) as a natural framework for cultivating emergent abilities such as question decomposition, query formulation, and evidence integration. However, most recent RL-based methods rely almost exclusively on outcome rewards (*e.g.*, exact match) (Fan et al., 2025; Jin et al., 2025), with the underlying assumption that optimizing for final answers will indirectly teach agents to search effectively. This assumption is questionable: as outcome-only supervision provides sparse, delayed feedback, it may suffer from the well-known credit assignment challenges (Alipov et al., 2021; Pignatelli et al., 2024). In other words, it is unclear whether outcome rewards alone can effectively promote intermediate search behaviors.

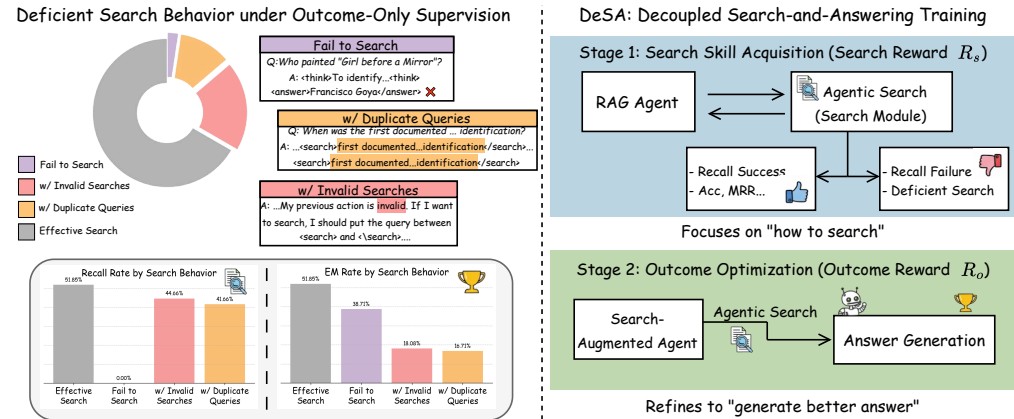

Figure 1: An overview of deficient search behaviors of agents trained with outcome-only supervision and an illustration of our DeSA (Decoupling Search-and-Answering). **(Left)** The results shown are collected from an agent trained solely based on a final-answer exact match (EM) reward with Qwen2.5-3B-Instruct as the backbone, and evaluated across seven QA datasets. This agent exhibits a variety of deficient search behaviors, including "Fail to Search", "w/ Invalid Searches", and "w/ Duplicate Queries". Compared to "Effective Search", these behaviors lead to significantly lower search recall and EM rate. **(Right)** DeSA decouples training into two stages to address these issues.

To investigate this, we first conduct a systematic behavior analysis of search agents trained with outcome-only rewards (Jin et al., 2025). In Section 4, we use Qwen2.5-3B/7B-Instruct (Qwen et al., 2025) as backbone models and evaluate their performance on seven question-answering benchmarks. As exemplified in Figure 1, the evaluation results of Qwen2.5-3B-Instruct reveals recurring deficiencies in search behavior: (1) **Fail to Search**: agents skip retrieval altogether and rely solely on parametric memory even when external knowledge is required; (2) **w/ Duplicate Queries**: agents issue identical queries across multiple action steps, wasting budget on redundant contents; (3) **w/ Invalid Searches**: the trajectories generated by agents contain malformed tool calls or meaningless queries that yield no useful information; and (4) **Mixtures** of these issues. These problematic behaviors not only impair search recall and efficiency but also ultimately lead to degraded answer accuracy.

Motivated by these findings, we propose **DeSA** (**De**coupling **S**earch and **A**nswering): a two-stage RL framework that explicitly separates the processes of learning to search and learning to answer. In Stage 1 (Search Skill Acquisition), agents focus purely on developing effective search behaviors, guided by a retrieval recall-based reward. In Stage 2 (Outcome Optimization), agents leverage outcome rewards to refine their ability to distill retrieved evidence into accurate answers. By decoupling these two distinct objectives, DeSA directly addresses the limitations of outcome-only optimization. Empirical experiments show that DeSA substantially improves both search quality and QA performance. For example, DeSA reduces the deficient search rate of Qwen2.5-3B-Instruct from 23.36% to 6.96% and increases search recall from 59.5% to 64.5%. Beyond search metrics, DeSA also delivers superior final accuracy, surpassing strong outcome-only baselines by 8.0% on the 3B model and 5.6% on the 7B model across seven diverse QA benchmarks. These results highlight the importance of explicitly decoupling search and answering, rather than relying on outcome-only reward.

Our main contributions in this work are as follows: (1) We conduct a detailed behavioral analysis of search agents trained with outcome rewards, systematically identifying and quantifying several deficient search patterns. Our analysis reveals the insufficiency of this common training paradigm for guiding agents to develop efficient search behaviors. (2) We introduce DeSA, a decoupled two-stage RL training framework that separates *Search Skill Acquisition* from *Outcome Optimization*. We demonstrate that DeSA significantly improves agent performance on a wide range of QA benchmarks compared to standard single-stage outcome-based training. (3) Through extensive ablation studies, we validate that our decoupled reward design outperforms single-stage (either with outcome reward or composite reward) and other two-stage variants.

## 2 RELATED WORK

### 2.1 RETRIEVAL-AUGMENTED GENERATION (RAG)

To address the factual hallucinations and static knowledge limitations of large language models (LLMs) (Huang et al., 2025), retrieval-augmented generation (RAG) (Gao et al., 2023; Asai et al., 2023) has emerged as an effective and widely-used technique that grounds large language models (LLMs) in external information. The concept was first introduced by Lewis et al. (2020), and since then, RAG has evolved from simple retrieve-then-generate pipelines to multi-step, iterative processes (Asai et al., 2024; Jin et al., 2024; Gao et al., 2024). These advances leverage the internal reasoning capabilities of LLMs, often elicited through techniques such as Chain-of-Thought (CoT) prompting (Wei et al., 2022) or post-training (Chan et al., 2024; Fang et al., 2024; Wei et al., 2025a), which enable models to break down complex problems and form multi-step plans. This reframes retrieval as a sequential decision-making problem and aligns the task with the broader paradigm of tool-using agents (Tang et al., 2023), where a search engine is an invocable tool. Although prompting methods (Yao et al., 2022) guide agents through in-context examples, training-based approaches (Schick et al., 2023) offer a more robust path to obtaining complex tool-use skills. However, high-quality tool-use trajectories are difficult to collect, highlighting the need for further research to improve these methods.

### 2.2 REINFORCEMENT LEARNING AND SEARCH AGENTS

Reinforcement learning (RL) (Kaelbling et al., 1996) is a machine learning paradigm in which an agent learns to make sequential decisions by interacting with an environment and receiving feedback in the form of rewards. RL methods encompass a variety of algorithms, from Policy Gradient (Schulman et al., 2017) and Proximal Policy Optimization (PPO) (Schulman et al., 2017), to simpler, preference-based approaches, including Direct Preference Optimization (DPO) (Rafailov et al., 2023) and its variants (Meng et al., 2024). GRPO (Shao et al., 2024) improves training efficiency by replacing PPO's value model with a rule-based reward function. Building on such advancements in RL algorithms, training LLMs with RL has become a popular paradigm (Zhu et al., 2025; DeepSeek-AI et al., 2025; Wei et al., 2025b). In the domain of search agents, works like Search-R1 (Jin et al., 2025), DeepResearcher (Zheng et al., 2025), and R1-Searcher (Song et al., 2025) train search agents end-to-end using an outcome-based reward. Sun et al. (2025a) use simulated searches to assist in training the agent's search capabilities. Despite the extensive practical work in this field (Fan et al., 2025; Jiang et al., 2025; Sha et al., 2025; Sun et al., 2025b; Zhao et al., 2025), few studies have investigated whether an outcome reward can effectively optimize search behavior. By default, they adopt an outcome-only reward with a questionable underlying assumption that optimizing for final answers also teaches agents to search effectively. This motivates our investigation into decoupling search from answer generation to better optimize both tasks.

## 3 PRELIMINARY

### 3.1 SEARCH-AUGMENTED AGENT

We adopt a general search-augmented agent workflow as used in many prior works (Jin et al., 2025; Song et al., 2025; Zheng et al., 2025), which is formulated as a sequential decision-making process. Given a user query $q$, the agent, equipped with an LLM policy $\pi_\theta$, interacts with a search engine over a series of steps to gather information before finally generating an answer. At each step $t$, the agent's state is defined by its history $H_t$, which contains the initial query and all preceding actions and observations: $H_t = (q, a_0, d_0, a_1, d_1, \ldots, a_{t-1}, d_{t-1})$. Based on this history, the agent is encouraged to think and reason over retrieved information (Wei et al., 2022; Jin et al., 2025), and then generate the next action: $a_t \sim \pi_\theta(\cdot|H_t)$. The possible actions are:

- `search`: Issues a search with $\text{query}_t$ to the search engine environment. The environment returns a set of top-ranked documents $d_t$ based on their relevance scores, which are then appended to the history. The new history becomes $H_{t+1} = H_t \oplus (a_t, d_t)$, where $\oplus$ denotes concatenation.

- `answer`: Concludes the search process and outputs the final answer to the user's question. This is a terminal action that ends the interactive process between the search agent and the environment.

Table 1: Prompt template for our search agent.

---

Answer the given question. You must conduct reasoning inside `<think>` and `</think>` first every time you get new information. After reasoning, if you find you lack some knowledge, you can call a search engine by `<search>` query `</search>`, and it will return the top searched results between `<information>` and `</information>`. You can search as many times as you want. If you find no further external knowledge needed, you can directly provide the answer inside `<answer>` and `</answer>` without detailed illustrations. For example, `<answer>` xxx `</answer>`. Question: question.

---

The formulation highlights a strong *sequential* dependence: the final answer is contingent upon the accumulated information, and each search action is chosen given the outcomes of preceding steps, highlighting the importance of search quality in the overall workflow. Following previous works (Jin et al., 2025), we adopt a commonly used prompt template for our search agent, as shown in Table 1.

## 3.2 Reinforcement Learning Algorithms

**Group Relative Policy Optimization (GRPO)** (Shao et al., 2024) is a reinforcement learning algorithm designed to optimize a policy without requiring an explicit value model, thereby reducing the computational and memory costs during large-scale RL training. Rather than estimating values with a separate value model, GRPO leverages a group of sampled outputs and computes a relative advantage for each response based typically on a verifiable reward function. Specifically, GRPO optimizes the following objective:

$$\mathcal{J}_{\text{GRPO}}(\theta) = \mathbb{E}_{x \sim \mathcal{D}, \{y_i\}_{i=1}^{G} \sim \pi_{\theta_{\text{old}}}(\cdot|x)} \left[ \frac{1}{G} \sum_{i=1}^{G} \frac{1}{|y_i|} \sum_{t=1}^{|y_i|} \min \left( w_{i,t}(\theta)\hat{A}_{i,t}, \text{clip}(w_{i,t}(\theta), 1-\epsilon, 1+\epsilon)\hat{A}_{i,t} \right) \right],$$

where $G$ is the number of generated responses to each query $x$ (i.e., the group size), and the importance ratio $w_{i,t}(\theta)$ and advantage $\hat{A}_{i,t}$ of token $y_{i,t}$ are defined as:

$$w_{i,t}(\theta) = \frac{\pi_\theta(y_{i,t}|x, y_{i,<t})}{\pi_{\theta_{\text{old}}}(y_{i,t}|x, y_{i,<t})}, \quad \hat{A}_{i,t} = \frac{r(x, y_i) - \text{mean}\left(\{r(x, y_j)\}_{j=1}^{G}\right)}{\text{std}\left(\{r(x, y_j)\}_{j=1}^{G}\right)},$$

all tokens in response $y_i$ share the same advantage $\hat{A}_i$. Besides, GRPO directly incorporates the KL divergence term for regularization into its objective, which contributes to training stability.

## 3.3 Outcome Reward

Following the standard Reinforcement Learning with Verifiable Rewards (RLVR) paradigm often employed with GRPO, previous search agent frameworks mainly use outcome reward to optimize agents' behavior. One of the representatives is Exact Match (EM) Reward:

**Exact Match (EM) Reward** The Exact Match (EM) reward is a binary signal that evaluates the correctness of the agent's final answer $a$. It is formally defined as:

$$R_{\text{EM}}(a, \mathcal{A}) = \begin{cases} 1 & \text{if } \exists a^* \in \mathcal{A} \text{ s.t. } \text{Normalized}(a) = \text{Normalized}(a^*) \\ 0 & \text{otherwise} \end{cases},$$

where $a$ is the agent's generated final answer, $\mathcal{A}$ is the set of ground-truth candidate answers, and Normalized($\cdot$) refers to a standardization function (*e.g.*, lowercasing, removing punctuation). This reward directly supervises the accuracy of the agent's final output, making it suitable for optimizing the agent's document denoising and accurate answer generation capability.

## 4 Search-behavior Analysis

### 4.1 Deficient Search-behavior Definitions

The central assumption of using an outcome-based reward (*e.g.*, exact-match accuracy) for training search agents is that optimizing the final answer will implicitly guide the agent's behavior, enabling

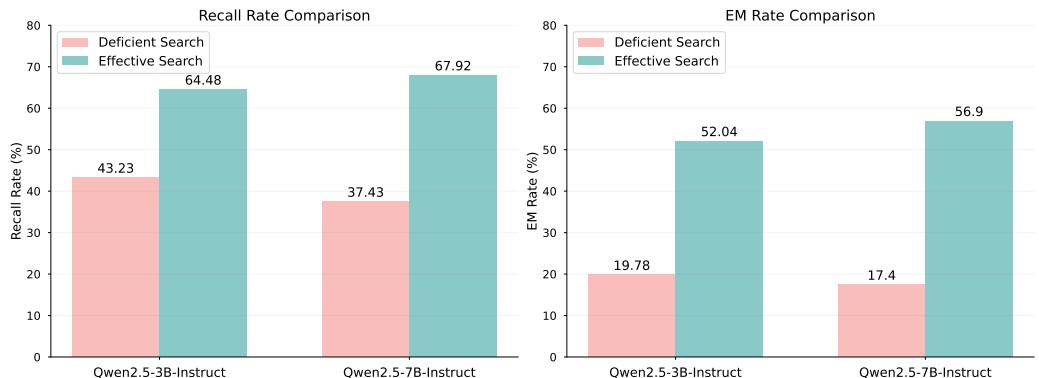

Figure 2: **Impact of deficient search behaviors on agent performance.** Both recall rate (left) and Exact Match (EM) rate (right) are significantly lower for trajectories exhibiting deficient behaviors compared to those with only effective behaviors.

it to become an effective searcher. To investigate this assumption, it is necessary to systematically examine the search behaviors that arise under such training. In this section, we highlight several deficient behaviors that emerge when agents are trained solely with outcome rewards. These behaviors serve as clear indicators of failure, revealing inefficiencies and ineffectiveness in the search process that ultimately hinder the generation of accurate final answers:

- **No Search:** The agent answers directly from its internal knowledge without using the search tool, even when the question requires external or up-to-date information. This often leads to factually incorrect or hallucinated answers.

- **w/ Duplicate Queries:** The agent repeatedly issues the same search query within a whole interaction. This is an inefficient strategy that wastes resources by retrieving redundant information without making progress.

- **w/ Invalid Searches:** The agent generates at least one invalid search action, such as mismatched or incomplete tags (*e.g.*, `<search>query/search`), or the query is meaningless (*e.g.*, blank space or just punctuations). Such actions result in a failed tool call without returning useful new information, wasting a step in the process.

## 4.2 ANALYSIS

To investigate the effectiveness of EM reward in optimizing search behaviors, we train search agents with EM reward on NaturalQuestions (Kwiatkowski et al., 2019) and HotpotQA (Yang et al., 2018), using Qwen2.5-3B-Instruct and Qwen2.5-7B-Instruct (Qwen et al., 2025) as backbones. We then evaluate their performance and inspected their search behavior on seven QA evaluation benchmarks (NaturalQuestions (Kwiatkowski et al., 2019), TriviaQA (Joshi et al., 2017), PopQA (Mallen et al., 2023), HotpotQA (Yang et al., 2018), 2WikiMultiHopQA (Ho et al., 2020), Musique (Trivedi et al., 2022), and Bamboogle (Press et al., 2023)), all the metrics are aggregated over these benchmarks. We define two goals for the search agent and examine its accomplishment under different search behaviors: 1. **Recall Success**: The retrieved information contains the correct answer. 2. **Answer Correctness (EM)**: The generated answer exactly matches a reference (or ground-truth) answer.

To validate whether **deficient search behaviors will decrease the search agent's performance**, we first divide all interaction trajectories into two distinct groups: those containing at least one of the deficient behaviors, and those with none. A clear performance gap is observed between these two groups, as shown in Figure 2: For the Qwen2.5-3B agent, flawed trajectories have a significantly lower average recall rate **(43.23% vs. 64.48%)**. This trend holds for the Qwen2.5-7B agent **(37.43% vs 67.92%)**. Similarly, the final EM rate on answer generation **dropped by 32.26% (19.78% vs. 52.04%)** for the 3B model and **39.5% (17.4% vs. 56.90%)** for the 7B model. To further dissect the nature of these failures, we analyzed the composition of all cases where the agent failed to recall the answer. As illustrated in Figure 3, our findings are: for the Qwen2.5-3B model, **33.3%** of all recall failures exhibited at least one of the deficient behaviors. This issue holds for the larger Qwen2.5-7B model, where the proportion is 19.58%. More seriously, the "Effective Search" behavior only denotes that it is clean-formed without collapsed patterns; there is still room to make

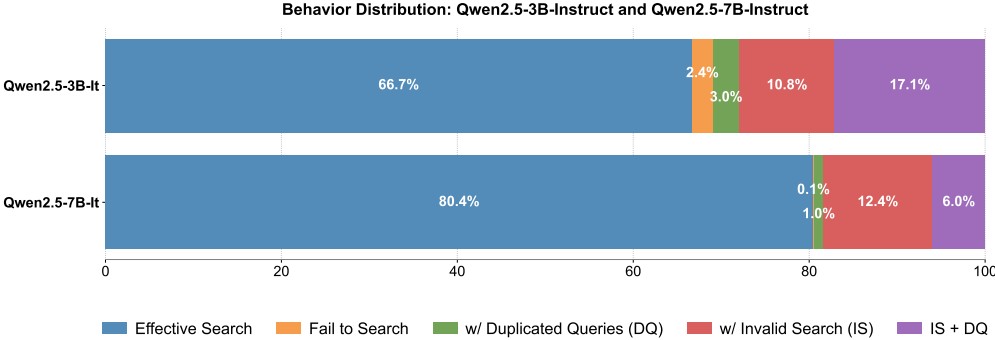

Figure 3: **Deficient Search Behaviors in Recall-Failure Cases.** This figure displays the distribution of the three defined deficient search behaviors, as well as their combinations, within all search trajectories that failed to recall the ground-truth answer.

it "Efficient" and achieve a higher recall rate. All of our analysis results provide strong evidence that **training with EM reward alone does not effectively translate the outcome reward signal into the optimization of intermediate search actions**, leading to frequent and predictable failure modes or inefficient searches.

## 5 METHOD

Our analysis shows that relying solely on an outcome-based EM reward does not effectively guide the intermediate search steps, leading to deficient and inefficient behaviors that reduce both recall and final EM accuracy. We therefore reconsider the search agent's sequentially dependent workflow and argue that successful search is a critical prerequisite for producing well-grounded answers. Building on this insight, we introduce DeSA (Algorithm 1), a two-stage decoupled training framework that explicitly separates these two objectives and applies tailored supervision sequentially: first, we train the agent exclusively on search efficacy, and then we train it to generate accurate answers from the evidence it has learned to retrieve effectively.

### 5.1 STAGE 1: SEARCH SKILL ACQUISITION

Given that an agent cannot answer questions for which it lacks sufficient supporting evidence, our objective in the first training stage is to provide a reward based on the retrieved information. This encourages the agent to gather more useful information, enhancing its search efficiency and efficacy, and mitigating deficient and inefficient search behaviors. We use the **Recall Reward** ($R_{\text{recall}}$) as our main objective. This reward provides a direct signal indicating whether the necessary information to answer the question has been successfully retrieved.

**Recall Reward.** Let $\mathcal{C} = \{c_1, c_2, \ldots, c_k\}$ be the set of all information blocks retrieved by the agent across its $k$ interactions. We define the aggregated recalled information $I_{\text{recalled}} = \text{Aggregate}(\mathcal{C})$. The recall reward is then formally given by:

$$R_{\text{recall}}(I_{\text{recalled}}, \mathcal{A}) = \begin{cases} 1 & \text{if } \exists a^* \in \mathcal{A} \text{ s.t. } a^* \in I_{\text{recalled}} \\ 0 & \text{otherwise} \end{cases} \tag{1}$$

This reward specifically incentivizes the agent to strategically generate search queries and effectively utilize the search engine to find supporting evidence, acting as a crucial signal for improving its information-seeking behavior. We also explored alternative search rewards for Stage 1 optimization. These included: (1) a composite reward that combines $R_{\text{recall}}$ with a penalty term to discourage deficient behaviors like duplicate queries. (2) a more fine-grained 'retrieval accuracy' reward ($R_{\text{acc}}$) that measures the proportion of retrieved documents containing the answer. Through experiments, we find that the simple recall reward generally works better than alternative (and more complicated) search reward designs, as discussed in Section **??**.

---

**Algorithm 1** The DeSA Training Framework

---

**Require:** Pre-trained LLM $\mathcal{M}_\theta$, Search Environment $\mathcal{E}$, Training Data $\mathcal{T} = \{\langle q, \mathcal{A} \rangle\}$
 1: {**Stage 1: Search Skill Acquisition**}
 2: **for** each training step in Stage 1 **do**
 3:     Sample question-answer pair $\langle q, \mathcal{A} \rangle$ from $\mathcal{T}$
 4:     Agent $\mathcal{M}_\theta$ interacts with environment $\mathcal{E}$ given $q$ to retrieve a set of information blocks $\mathcal{C}$
 5:     Aggregate recalled information $I_{\text{recalled}} = \text{Aggregate}(\mathcal{C})$
 6:     Compute **Search Reward** $R_{\text{recall}}(I_{\text{recalled}}, \mathcal{A})$ using Eq. 1
 7:     Update parameters $\theta$ of agent $\mathcal{M}_\theta$ to maximize $R_{\text{recall}}$
 8: **end for**
 9:
10: {**Stage 2: Outcome Optimization**}
11: **for** each training step in Stage 2 **do**
12:     Sample question-answer pair $\langle q, \mathcal{A} \rangle$ from $\mathcal{T}$
13:     Agent $\mathcal{M}_\theta$ performs search and generates a final answer $a_{\text{pred}}$ for question $q$
14:     Compute **Outcome Reward (EM)** $R_{\text{EM}}(a_{\text{pred}}, \mathcal{A})$
15:     Update parameters $\theta$ of agent $\mathcal{M}_\theta$ to maximize $R_{\text{EM}}$
16: **end for**
17: **return** Trained Agent $\mathcal{M}_\theta$

---

## 5.2 STAGE 2: OUTCOME OPTIMIZATION

After establishing foundational search skills in Stage 1, the second stage shifts the optimization focus to the outcome. The primary goal is to train the agent to effectively translate retrieved information into correct answers by enhancing its ability to de-noise documents, synthesize evidence, and generate a precise final solution. For this purpose, we fine-tune the agent using the **outcome reward (Exact Match)** ($R_{\text{EM}}$).

By initializing this stage with the model checkpoint from Stage 1, we ensure that the agent builds upon its improved search capabilities while still maintaining reasonable behaviors. Stage 2 training also allows the agent to adapt its search behavior from exhaustive search to more accurate under the optimization pressure on the final answer generation task. This stage ensures the agent leverages the higher-quality context provided by its search skills and achieves superior downstream performance.

## 6 EXPERIMENTS

### 6.1 EXPERIMENTAL SETUP

**Setup**    To ensure a fair and direct comparison, we follow the implementation of Search-R1 (Jin et al., 2025) and keep our experimental settings closely aligned with it. We utilize Qwen2.5-3B-Instruct and Qwen2.5-7B-Instruct (Qwen et al., 2025) as the backbone large language models and GRPO as the RL algorithm for our agents. We follow previous work and integrate the loss masking for the retrieved tokens. For the training, we use a training dataset includes the training splits of Natural Questions (NQ) (Kwiatkowski et al., 2019) and HotpotQA (Yang et al., 2018). We use the 2018 Wikipedia as the knowledge corpus and E5 (Wang et al., 2022) as the retriever, which fetches the top 3 relevant passages for each search query.

**Datasets**    We evaluate our models on a comprehensive suite of seven question-answering benchmarks to assess both in-domain and out-of-domain performance. These datasets include: (1) General QA: NaturalQuestions, TriviaQA (Joshi et al., 2017), and PopQA (Mallen et al., 2023); and (2) Multi-Hop QA: HotpotQA, 2WikiMultiHopQA (Ho et al., 2020), Musique (Trivedi et al., 2022), and Bamboogle (Press et al., 2023). Following standard practice for these benchmarks, we use Exact Match (EM) as the primary evaluation metric.

**Baselines**    Our primary baseline is the single-stage training approach from Search-R1 (Jin et al., 2025), which uses only the EM reward. For a broader comparison, we also include baselines with

Table 2: Main results. The best performance is set in bold. $^{\dagger}$ / $^{*}$ represents in-domain/out-of-domain datasets.

| Methods | General QA | | | Multi-Hop QA | | | | Avg. |
|---|---|---|---|---|---|---|---|---|
| | NQ$^{\dagger}$ | TriviaQA$^{*}$ | PopQA$^{*}$ | HotpotQA$^{\dagger}$ | 2Wiki$^{*}$ | Musique$^{*}$ | Bamboogle$^{*}$ | |
| Qwen2.5-7b-Instruct | | | | | | | | |
| Direct Inference | 0.134 | 0.408 | 0.140 | 0.183 | 0.250 | 0.031 | 0.120 | 0.181 |
| CoT | 0.048 | 0.185 | 0.054 | 0.092 | 0.111 | 0.022 | 0.232 | 0.106 |
| IRCoT | 0.224 | 0.478 | 0.301 | 0.133 | 0.149 | 0.072 | 0.224 | 0.239 |
| Search-o1 | 0.151 | 0.443 | 0.131 | 0.187 | 0.176 | 0.058 | 0.296 | 0.206 |
| RAG | 0.349 | 0.585 | 0.392 | 0.299 | 0.235 | 0.058 | 0.208 | 0.304 |
| SFT | 0.318 | 0.354 | 0.121 | 0.217 | 0.259 | 0.066 | 0.112 | 0.207 |
| R1-base | 0.297 | 0.539 | 0.202 | 0.242 | 0.273 | 0.083 | 0.296 | 0.276 |
| R1-instruct | 0.270 | 0.537 | 0.199 | 0.237 | 0.292 | 0.072 | 0.293 | 0.271 |
| Rejection Sampling | 0.360 | 0.592 | 0.380 | 0.331 | 0.296 | 0.123 | 0.355 | 0.348 |
| Search-R1 | 0.429 | 0.623 | 0.427 | 0.386 | 0.346 | 0.161 | **0.400** | 0.396 |
| **DeSA (Ours)** | **0.468** | **0.631** | **0.440** | **0.424** | **0.374** | **0.197** | 0.395 | **0.418** |
| Qwen2.5-3b-Instruct | | | | | | | | |
| Direct Inference | 0.106 | 0.288 | 0.108 | 0.149 | 0.244 | 0.020 | 0.024 | 0.134 |
| CoT | 0.023 | 0.032 | 0.005 | 0.021 | 0.021 | 0.002 | 0.000 | 0.015 |
| IRCoT | 0.111 | 0.312 | 0.200 | 0.164 | 0.171 | 0.067 | 0.240 | 0.181 |
| Search-o1 | 0.238 | 0.472 | 0.262 | 0.221 | 0.218 | 0.054 | 0.320 | 0.255 |
| RAG | 0.348 | 0.544 | 0.387 | 0.255 | 0.226 | 0.047 | 0.080 | 0.270 |
| SFT | 0.249 | 0.292 | 0.104 | 0.186 | 0.248 | 0.044 | 0.112 | 0.176 |
| R1-base | 0.226 | 0.455 | 0.173 | 0.201 | 0.268 | 0.055 | 0.224 | 0.229 |
| R1-instruct | 0.210 | 0.449 | 0.171 | 0.208 | 0.275 | 0.060 | 0.192 | 0.224 |
| Rejection Sampling | 0.294 | 0.488 | 0.332 | 0.240 | 0.233 | 0.059 | 0.210 | 0.265 |
| Search-R1 | **0.397** | 0.565 | 0.391 | 0.331 | 0.310 | 0.124 | 0.232 | 0.336 |
| **DeSA (Ours)** | 0.375 | **0.575** | **0.397** | **0.352** | **0.363** | **0.134** | **0.347** | **0.363** |

other existing methods (*e.g.*, RAG, IRCoT (Trivedi et al., 2023), and SFT), citing the results reported in the Search-R1 paper.

## 6.2 MAIN RESULTS

**Overall Performance**   As presented in Table 2, our proposed DeSA significantly outperforms all baseline methods across both model sizes. For the larger Qwen2.5-7B-Instruct model, DeSA achieves the top average score of 0.418, representing a 5.6% relative improvement over the strong single-stage Search-R1 baseline (0.396). The performance gains are particularly notable on complex multi-hop question-answering tasks such as HotpotQA (+3.8 points) and Musique (+3.6 points), demonstrating DeSA's ability to facilitate more effective search strategies. The advantages of our decoupled pipeline are even more pronounced on the smaller Qwen2.5-3B-Instruct model. DeSA achieves an average score of 0.363, outperforming Search-R1 (0.336) by 8.0%. The gains are especially significant on challenging out-of-domain datasets, including a remarkable 11.5-point absolute improvement on Bamboogle (0.347 vs. 0.232) and a 5.3-point gain on

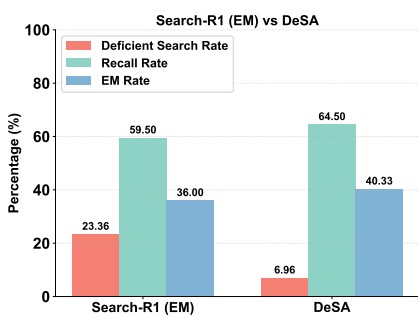

Figure 4: Final performance comparison of DeSA vs. the single-stage Search-R1 baseline on the 3B model.

2WikiMultiHopQA. The fact that the performance gap widens on the smaller model suggests that our two-stage approach provides a more crucial and effective learning signal, compensating for the model's reduced capacity by explicitly rewarding the acquisition of search skills before optimizing the outcome. To verify the reliability of these results, we conducted Micro-Average Z-tests across the full evaluation set ($N = 51,713$). DeSA demonstrates highly statistically significant improvements over the baseline ($p < 10^{-9}$), confirming that the gains are robust and practically meaningful.

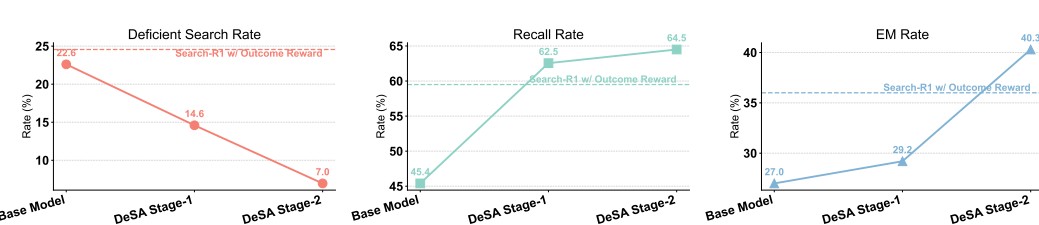

Figure 5: **Performance breakdown across DeSA's two stages**, compared with Search-R1 baseline.

Table 3: Ablation study on the Qwen2.5-3B-Instruct model. We compare DeSA against Stage 1 reward variations and various Single-stage (End-to-End) baselines. † and * denote datasets from **?** and Press et al. (2023), respectively.

| Methods | General QA | | | Multi-Hop QA | | | | |
|---|---|---|---|---|---|---|---|---|
| | NQ† | TriviaQA* | PopQA* | HotpotQA† | 2Wiki* | Musique* | Bamboogle* | Avg. |
| **DeSA (Ours)** | 0.375 | 0.575 | 0.397 | **0.352** | **0.363** | **0.134** | **0.347** | **0.363** |
| *Stage 1 Reward Analysis* | | | | | | | | |
| w/ $R_{recall}$+$R_{penalty}$ | 0.370 | 0.567 | 0.397 | 0.337 | 0.342 | 0.112 | 0.306 | 0.347 |
| w/ $R_{acc}$ | 0.374 | 0.577 | **0.404** | 0.347 | 0.346 | 0.125 | 0.306 | 0.354 |
| *Single-stage Baselines* | | | | | | | | |
| $0.2R_{recall} + 0.8R_{EM}$ | 0.382 | 0.573 | 0.401 | 0.349 | 0.337 | 0.130 | 0.323 | 0.356 |
| $0.5R_{recall} + 0.5R_{EM}$ | 0.386 | 0.563 | 0.383 | 0.348 | 0.330 | 0.132 | 0.307 | 0.350 |
| $0.8R_{recall} + 0.2R_{EM}$ | 0.329 | 0.540 | 0.375 | 0.306 | 0.304 | 0.098 | 0.218 | 0.310 |
| EM + Penalties | **0.387** | **0.578** | **0.404** | 0.345 | 0.356 | 0.121 | 0.266 | 0.351 |

**Evolution of Search Behavior**   As illustrated in Figure 4, on the 3B model, our DeSA method achieves a significantly lower deficient search rate compared to Search-R1, which is trained solely with an EM reward (6.96 vs. 23.36). Benefiting from this higher-quality search, DeSA also obtains considerably higher recall and EM scores (64.5 vs. 59.5 and 40.33 vs. 36.00, respectively). Figure 5 explains how our two-stage training affects the performance. After Stage 1 (Search Skill Acquisition), the agent's deficient search rate drops significantly below the Search-R1 baseline (14.60 vs. 23.36), while its recall already surpasses it (62.55 vs. 59.50). It demonstrates that the agent can already do efficient search after Stage 1. Following this, Stage 2 (Outcome Optimization) brings a sharp increase in the EM rate (from 29.2 to 40.3). Concurrently, the agent refines its search strategy from a slightly exhaustive, recall-focused approach to a more precise one, causing the deficient search rate to fall further to 6.96 and the recall to modestly increase to 64.50. This progression validates the effectiveness of our two-stage training methodology.

## 6.3 ABLATION STUDY

### 6.3.1 REWARD DESIGN FOR STAGE 1

We conduct experiments to analyze the impact of different reward formulations in Stage 1, using our primary DeSA model (trained with only $R_{recall}$) as the baseline for comparison. First, we investigate a composite reward, $R_{recall} + R_{penalty}$, which adds a penalty of -0.2 for each rollout if it includes a deficient search behavior (*e.g.*, duplicate or invalid queries). Second, we evaluate a more fine-grained signal, Retrieval Accuracy ($R_{acc}$), which we define as the proportion of retrieved documents containing a ground-truth answer. As shown in Table 3, manually adding the behavior penalty decreases final performance; while it encourages more stable behavior (more details in Appendix B.3), it also appears to restrict the agent's ability to maximize recall. The fine-grained Retrieval Accuracy reward improves performance on General QA tasks but decreases on complex Multi-Hop QA, resulting in a lower overall score.

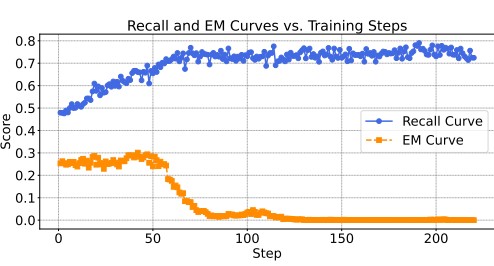 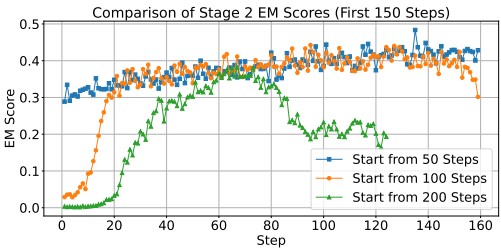

(a) Stage 1 Recall and EM curves.      (b) Stage 2 EM curves from different checkpoints.

Figure 6: **Analysis of the optimal transition point from Stage 1 to Stage 2.**

### 6.3.2 SINGLE-STAGE VS. TWO-STAGE TRAINING

To validate our central hypothesis that decoupling search and answering is beneficial, we compare our proposed two-stage method against some single-stage alternatives. The baseline is trained from the initial model using a reward function that linearly combines the recall and EM signals ($R = w \times R_{\text{recall}} + (1-w) \times R_{\text{EM}}$). We explore varying weights $w \in \{0.2, 0.5, 0.8\}$. As shown in Table 3, our two-stage approach outperforms even the best-performing single-stage variant ($w = 0.2$). This suggests that mixing the search reward (recall) and outcome reward (EM) creates confusing optimization pressures, making it difficult for the agent to learn effectively. In contrast, our decoupled pipeline allows the agent to first master the search process before focusing on answer generation, leading to a more robust agent policy. To further verify the necessity of our proposed $R_{\text{recall}}$ training, we also tested an end-to-end baseline using EM with behavior penalties. While effective on General QA, it lags significantly on complex benchmarks (e.g., -8.1% on Bamboogle), confirming that explicit search supervision is essential for learning effective multi-hop exploration.

### 6.3.3 TRANSITION POINT OF THE TWO-STAGE TRAINING

The effectiveness of DeSA depends on selecting an appropriate transition point from Stage 1 to Stage 2. As illustrated in Figure 6a, continued training in Stage 1 produces a characteristic pattern: the exact-match (EM) score rises to an early peak (around 50 steps in our setting) and then drops abruptly, even though the recall metric continues to improve. This divergence indicates that the model has started to exploit the recall-based reward at the expense of generating correct final answers. We therefore use the last pre-collapse checkpoint (the point just before the EM curve turns downward) as a practical marker for transitioning. To test this guideline, we initiate Stage 2 training from checkpoints at 50, 100, and 200 steps. As shown in Figure 6b, we observe that when Stage 2 begins after the collapse has already started, the agent requires substantially more training to recover its question-answering behavior, and the second stage becomes less stable. This demonstrates that identifying the transition point via the EM curve provides a concrete and effective criterion for moving between stages.

## 7 CONCLUSION

In our work, we identify and analyze the inefficacy of using outcome-only rewards to optimize the search behavior of agents. We propose DeSA (Decoupling Search-and-Answering), a two-stage training pipeline that separates search skill acquisition from answer generation optimization. Comprehensive experimental evidence demonstrates that DeSA achieves superior search quality (in terms of higher recall and fewer deficient behaviors) and significantly improves performance on QA datasets compared to baselines. Our work represents a re-examination of the current agentic RL training paradigm, emphasizing the value of process-based rewards.

For future work, we plan to explore more advanced process-based rewards for Stage 1 training, such as using a dedicated reward model to evaluate the agent's search behavior. Furthermore, we aim to extend the principle of our DeSA framework to broader agentic tasks beyond question-answering. We believe this principle could also be effective in domains such as code generation and long-context understanding, within both single-agent and multi-agent settings.

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

# A    TRAINING CONFIGURATION

For our GRPO training, we use the same training configuration for the two stages. We set the policy LLM learning rate to $1e^{-6}$ and sample 5 responses per prompt for advantage estimation. The model is trained for a total of 1005 steps, with a learning rate warm-up ratio of 0.285.

Training is conducted on a single node with 4 GPUs. We use a total training batch size of 512. For the policy update, the mini-batch size is 256 and the micro-batch size is 32. The maximum sequence length is set to 4,096 tokens, with a maximum response length of 500 and a maximum length of 500 tokens for retrieved content. To optimize memory usage, we enable gradient checkpointing and use Fully Sharded Data Parallel (FSDP) with CPU offloading for parameters, gradients, and the optimizer state.

For efficient rollouts, we use the vLLM engine with a sampling temperature of 1.0. The KL divergence regularization coefficient is set to 0.001. During interactions, the maximum number of turns is set to 4, and we retrieve the top 3 passages for each search query. Model checkpoints are saved every 50 steps, and the final checkpoint before collapse is used for evaluation.

# B    SUPPLEMENTARY EXPERIMENTAL RESULTS AND ANALYSIS

## B.1    DEFICIENT SEARCH BEHAVIOR DISTRIBUTION BREAKDOWN

To demonstrate the systematic improvements achieved by DeSA, we present a fine-grained analysis of failure modes. Table 4 compares the deficient search behavior rates of the baseline (Search-R1) versus DeSA (using Qwen2.5-3B-Instruct backbone) across the total evaluation set ($N = 51,712$). As illustrated in the quantitative results, DeSA significantly suppresses all types of deficient behaviors. Notably, **Invalid Searches** are reduced by nearly 90%, and **Duplicate Queries** are reduced by over 60%. This reduction highlights the effectiveness of our method in regularizing the search process and minimizing inefficient exploration.

Table 4: Comparison of deficient search behavior rates between the baseline (Search-R1) and DeSA across the full evaluation set. DeSA achieves a 70.2% reduction in total deficient behaviors.

| Deficiency Category | Search-R1 (Baseline) | DeSA (Ours) | Reduction Rate |
|---|---|---|---|
| Fail to Search | 0.78% | 0.11% | ↓ 85.9% |
| Duplicate Queries | 13.88% | 5.30% | ↓ 61.8% |
| Invalid Search | 20.34% | 2.09% | ↓ 89.7% |
| **Total Deficient Rate** | **23.36%** | **6.96%** | **↓ 70.2%** |

*Note: Percentages represent the proportion of trajectories exhibiting the specific failure mode relative to the full dataset. The column sum exceeds the Total Deficient Rate because a single trajectory may contain multiple types of errors (e.g., both duplicate and invalid queries).*

## B.2    EFFECTIVENESS ON BASE MODELS (QWEN2.5-3B-BASE)

To validate the effectiveness of DeSA beyond instruction-tuned models, we conducted additional experiments using **Qwen2.5-3B** (Base) as the backbone. As presented in Table 5, DeSA achieves substantial gains on the base model, significantly outperforming the Search-R1 baseline. Specifically, DeSA raises the average performance from **0.303** to **0.396**, representing a relative improvement of **30.7%**. This improvement margin is even larger than that observed on Instruct models, suggesting that base models benefit more from our explicit search training. Notably, the performance of the 3B DeSA (Base) comes close to the 7B Search-R1 baseline (0.396 vs. 0.396), demonstrating that our decoupled training framework is highly effective at instilling robust search and reasoning capabilities into base models from scratch.

Table 5: Performance comparison using **Qwen2.5-3B-Base** as the backbone. DeSA achieves a 30.7% relative improvement over the Search-R1 baseline.

| Method | NQ | TriviaQA | PopQA | HotpotQA | 2Wiki | Musique | Bamboogle | Avg. |
|---|---|---|---|---|---|---|---|---|
| Search-R1 (Base) | 0.406 | 0.587 | 0.435 | 0.284 | 0.273 | 0.049 | 0.088 | 0.303 |
| **DeSA (Base)** | **0.470** | **0.625** | **0.448** | **0.396** | **0.365** | **0.145** | **0.323** | **0.396** |

B.3 SUPPLEMENTARY RESULTS FOR STAGE 1'S REWARD DESIGN

Compared to Stage-1 training with $R_{\text{recall}}$ only, $R_{\text{recall}} + R_{\text{penalty}}$ results in a lower deficient search rate (5.09% vs 14.6%) on the evaluation sets. However, as shown in Figure 7, after about 40 steps, its mean training recall lags behind $R_{\text{recall}}$, which suggests that this somehow restricts the agent from developing its search efficiency.

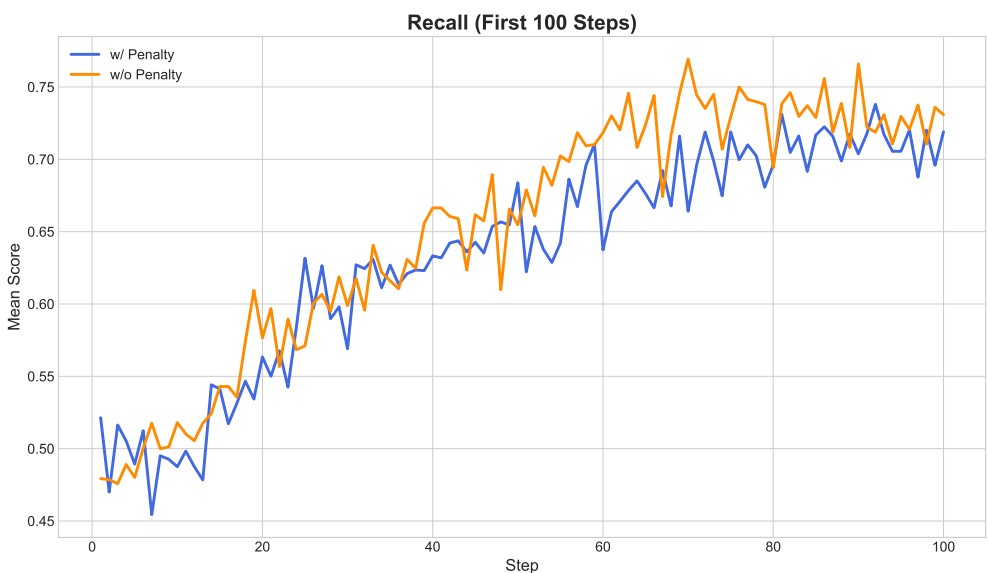

Figure 7: **Recall Comparison of First Stage.** The blue line is trained with $R_{\text{recall}} + R_{\text{penalty}}$ and the orange line is trained with $R_{\text{recall}}$ only.

B.4 STAGE-1 TRAINING DYNAMICS: ANALYSIS OF REWARD HACKING

We observed a distinct Reward Hacking phenomenon when training extends beyond the optimal point (approx. Step 50 in our experiments).

**Why EM Crashes** In Stage 1, the agent is optimized solely based on retrieval recall ($R_{\text{recall}}$). As the policy converges towards maximizing this specific reward, the agent discovers a "shortcut": it can further maximize recall by dedicating its entire action budget ($N = 4$ turns) exclusively to search actions. By issuing as many queries as possible, it increases the probability of hitting relevant documents. Consequently, the model ceases to generate the final answer structure (e.g., the `<answer>` tags) entirely. It effectively views the answering phase as a "waste" of actions that does not contribute to the retrieval reward. This behavior causes the Exact Match (EM) score to crash to near zero, despite recall continuing to rise.

**Justification for Early Stopping** This observation provides a principled rationale for our stage transition strategy. The point of "peak EM before collapse" marks the onset of an **objective shift**, where the agent begins to over-optimize the sub-objective (recall) at the expense of general reasoning coherence. Stopping at this peak serves as a necessary Early Stopping mechanism to prevent the

agent from overfitting to the proxy task, ensuring it retains the capability to generate answers before entering Stage 2.

## C    QUALITATIVE ANALYSIS AND CASE STUDIES

To better understand the performance gains achieved by DeSA compared to the Search-R1 baseline, as well as the limitations that persist, we conducted a qualitative analysis of interaction trajectories. All trajectories are collected using the Qwen2.5-3B-Instruct backbone.

### C.1    DeSA IMPROVEMENTS: MITIGATING DEFICIENT BEHAVIORS AND ENHANCING REASONING

Our analysis reveals that DeSA significantly outperforms the baseline by overcoming specific deficient patterns common in outcome-reward trained agents. We highlight three primary improvement categories: (1) correcting *Fail-to-Search* and parametric hallucinations, (2) breaking *Duplicate Query Loops*, and (3) achieving *Precise Entity Extraction* in complex contexts.

#### C.1.1    MITIGATION OF "FAIL TO SEARCH" (OVERCONFIDENCE)

A common failure mode in the baseline is relying on incorrect internal parametric memory without verifying via search. DeSA's training curriculum forces the agent to verify information, successfully correcting these hallucinations.

---

**Case 1: Mitigation of "Fail to Search"**

**Question:** *What was the first video game for Nintendo?*

---

**Baseline (Search-R1) Trajectory:**

*Thought:* "...I check basic knowledge records... Nintendo released their first domestically-produced game... Donkey Kong..."

*Action:* [No Search Performed]

*Output:* "Donkey Kong"  ✗  *(Incorrect: Donkey Kong was 1981; EVR Race was 1975)*

---

**DeSA Trajectory:**

*Action:* `<search>first video game released by Nintendo</search>`

*Observation:* [Docs mention EVR Race (1975), Color TV-Game series...]

*Reasoning:* "...The search results indicate EVR Race preceded Donkey Kong..."

*Output:* "EVR Race"  ✓

---

Table 6: The baseline relies on parametric memory and hallucinates. DeSA engages the search tool to retrieve the correct historical fact.

#### C.1.2    BREAKING DUPLICATE QUERY LOOPS

The baseline agent sometimes ignores retrieved evidence and generates repeated, identical search queries. DeSA mitigates this issue.

#### C.1.3    PRECISE ENTITY EXTRACTION IN COMPLEX CONTEXTS

DeSA demonstrates superior multi-step reasoning when documents contain multiple related entities (e.g., distinguishing between a lead actor and a specific voice role). The baseline selects the most prominent entity, whereas DeSA filters context more accurately.

### C.2    FAILURE MODE ANALYSIS

To fairly assess DeSA's limitations, we conducted a qualitative analysis of error cases. We categorize the primary failure modes into two distinct types: temporal misalignment between the query and corpus, and fundamental gaps in reasoning capabilities.

---

**Case 2: Mitigation of Duplicate Queries**

**Question:** *What year did Seven Nation Army come out?*

---

**Baseline (Search-R1) Trajectory:**

*Thought:* "...The hit is most famously associated with the rock band Dire Straits [Hallucination]..."

*Step 1:* `<search>`when did dire stattres seven nation army come out`</search>`

*Observation:* [Docs: "Seven Nation Army" is a song by The White Stripes... released in 2003...]

*Step 2:* `<search>`when did dire stattres seven nation army come out`</search>` (Duplicate Loop)

*Output:* "1985" ✗ *(Hallucinated date based on Dire Straits era)*

---

**DeSA Trajectory:**

*Step 1:* `<search>`when did dire stattres seven nation army come out`</search>`

*Observation:* [Docs: "Seven Nation Army"... by The White Stripes... March 2003...]

*Reasoning:* "...From the findings... it's clear that the song was by The White Stripes, released in 2003... my [initial] misconception needs reevaluation..."

*Output:* "2003" ✓

---

Table 7: The baseline ignores search results and issues duplicate queries. DeSA corrects its premise based on the retrieved evidence.

---

**Case 3: Precise Entity Extraction**

**Question:** *Who does the voice of the beast in the new movie?*

---

**Baseline (Search-R1) Response:**

*Step 1:* `<search>`Who does the voice of the beast in the new movie?`</search>`

*Observation:* [Docs mention Beauty and the Beast (2017) stars Emma Watson and Dan Stevens...]

*Output:* "Emma Watson" ✗

*(Error: Selected the most prominent name (Belle) instead of the Beast.)*

---

**DeSA Response:**

*Step 1:* `<search>`Who does the voice of the beast in the new movie?`</search>`

*Observation:* [...The film features an ensemble cast that includes Emma Watson and Dan Stevens as the eponymous characters...]

*Step 2:* `<search>`Who did voice the Beast in the 2017 Disney movie?`</search>`

*Observation:* [...cast members including Emma Watson, Dan Stevens... Watson has a soprano voice...]

*Reasoning:* "...Walt Disney Pictures casts Emma Watson to voice Beast [Incorrect assumption]... wait, the film features Emma Watson and Dan Stevens as the eponymous characters... checking specific role..."

*Output:* "Dan Stevens" ✓

---

Table 8: DeSA successfully engages in iterative verification to distinguish the specific voice actor from the lead actress.

### C.2.1 TEMPORAL MISMATCH (QUESTION-KNOWLEDGE ALIGNMENT)

A significant challenge in Open-Domain QA is the discrepancy between the static retrieval corpus and questions implying dynamic or future events. As shown in Case 4, the question asks for a "next" event. The model fails to align the question's current temporal intent with the static timestamp of the retrieved documents, leading to a hallucinated date.

---

**Case Study 4: Temporal Mismatch**

**Question:** *When is the next Deadpool movie being released?*

---

**Retrieved Context:** [Documents from ∼2018 discuss the release of *Deadpool 2* in May 2018 and mention vague future plans.]

**DeSA Output:** "May 18, 2020"  ✗

**Ground Truth:** "May 18, 2018" (Referring to Deadpool 2, relative to the dataset timeframe).

**Analysis:** This error stems from a **temporal alignment failure**. The model attempts to extrapolate a "future" date (2020) to satisfy the word "next," ignoring that within the static corpus context, the correct answer was the 2018 release.

---

### C.2.2 REASONING & COMPREHENSION GAPS

The second category highlights limitations in the backbone model's reading comprehension and commonsense reasoning. In these cases, the retrieval is accurate, but the model fails to interpret the specific **semantic constraint** (e.g., answer type) imposed by the question phrasing.

---

**Case Study 5: Semantic Misinterpretation**

**Question:** *The south west wind blows across Nigeria between?*

---

**Retrieved Context:** "...an airmass originating from the south Atlantic Ocean, locally known as the south west wind... invades the country from February... reaching the northern part in June..."

**DeSA Reasoning:** "...The Tropical Maritime airmass (MT), a local known as South West Wind... further attests the flow is indeed from the southwest direction originating from south Atlantic Ocean..."

**DeSA Output:** "South Atlantic Ocean"  ✗

**Ground Truth:** "till September" (Temporal Duration).

**Analysis:** The model successfully retrieved the paragraph containing both the origin (ocean) and the timing (months). However, it suffered a **reasoning failure**: it misinterpreted the preposition "between" as asking for a spatial location/origin rather than a temporal interval, incorrectly extracting the location instead of the time.

---

## D DATASET DISTRIBUTION DETAILS

In this section, we provide the detailed statistics of the training and evaluation datasets used in our experiments, categorized by their reasoning complexity (Single-hop vs. Multi-hop).

### D.1 TRAINING SET STATISTICS

The training set is composed of Natural Questions (NQ) and HotpotQA, balancing single-hop retrieval skills with multi-hop reasoning capabilities. The distribution is shown in Table 9.

Table 9: Distribution of the Training Dataset.

| Dataset | Query Type | Sample Count |
|---|---|---|
| Natural Questions (NQ) | Single-hop | 79,168 |
| HotpotQA | Multi-hop | 90,447 |
| **Total** | | **169,615** |

### D.2 EVALUATION SET STATISTICS

To assess the model's generalization capabilities, we constructed a comprehensive evaluation set containing 51,713 samples across seven benchmarks. This set covers both simple fact retrieval (Single-hop) and complex reasoning (Multi-hop) tasks, as detailed in Table 10.

Table 10: Distribution of the Evaluation (Test) Dataset.

| Dataset | Query Type | Sample Count | Proportion (%) |
|---|---|---|---|
| PopQA | Single-hop | 14,267 | 27.6% |
| TriviaQA | Single-hop | 11,313 | 21.9% |
| Natural Questions (NQ) | Single-hop | 3,610 | 7.0% |
| 2WikiMultiHopQA | Multi-hop | 12,576 | 24.3% |
| HotpotQA | Multi-hop | 7,405 | 14.3% |
| Musique | Multi-hop | 2,417 | 4.7% |
| Bamboogle | Multi-hop | 125 | 0.2% |
| **Total** | | **51,713** | **100.0%** |

# E   LARGE LANGUAGE MODELS USAGE

During the preparation of this manuscript, we utilized Large Language Models (LLMs) for assistance. The primary applications included refining language and improving clarity, as well as generating Python code to create the data visualizations presented in the figures. All content, including text and figures, was carefully reviewed and edited by the authors, who take full responsibility for the final version of this work.

