# OpenReview forum: "Beyond Outcome Reward: Decoupling Search and Answering Improves LLM Agents"
_ICLR.cc/2026/Conference — Submitted to ICLR 2026_

### Official Review · Reviewer_8b1L · 2025-10-26

**Soundness:** 2
**Presentation:** 2
**Contribution:** 2
**Rating:** 4
**Confidence:** 4

**Summary:**

Through systematic analysis, the authors demonstrate that outcome-only training leads to multiple deficient search behaviors, including agents failing to invoke tools, issuing invalid queries, and performing redundant searches, ultimately degrading final answer quality.
To remedy this, the authors propose DeSA (Decoupling Search-and-Answering), a two-stage training framework using Reinforcement Learning (RL). Stage 1 (Search Skill Acquisition) uses a recall-based reward (R recall) to explicitly train the agent to gather useful information and mitigate inefficient behaviors. Stage 2 (Outcome Optimization) fine-tunes the agent using the standard Exact Match (EM) reward to improve document denoising, evidence synthesis, and accurate answer generation.

**Strengths:**

- DeSA achieves superior overall accuracy compared to strong baselines across a comprehensive suite of seven question-answering benchmarks. The benchmark suite includes both General QA and complex Multi-Hop QA.
- The proposed DeSA framework is intuitive, simple, and easy to reimplement. Its success, despite its simplicity, is a significant plus.

**Weaknesses:**

- The R_recall reward (Eq 4) is ad hoc and appears insufficient for complex, multi-hop questions. The formulation gives a reward if any ground-truth answer is found in the context. For questions requiring multiple pieces of evidence, this reward signal is flawed—it would reward an agent for finding only one of two necessary facts, which is not enough to answer the question. This reward function also fails to distinguish between a single precise query and multiple redundant ones—yet the paper criticizes redundancy as a deficiency.
- The claim that “simple recall works better” (Section 6.3.1) is based on final EM, not search behavior quality. I feel there isn't enough ablation studies to draw to this conclusion here. A more nuanced reward might improve both.

**Questions:**

- How is the ground-truth answer set A (used for R_recall) defined, especially for multi-hop training data like HotpotQA? Is it just the final answer string, or does it include the intermediate supporting facts?
- The choice to switch stages at “peak EM before collapse” seems post-hoc. Is there a principled, automatic way to detect this transition point without monitoring EM?

---

> ### Author Response · Authors · 2025-11-26
>
> We sincerely thank the reviewer for the insightful comments. We appreciate the opportunity to clarify our design choices regarding the reward formulation and to provide a more detailed analysis of search behaviors.
>
> ---
>
> > **[W1 & Q1]:** The validity of $R_{recall}$ for multi-hop questions and the definition of the Ground-Truth set $\mathcal{A}$.
>
> **[A1]:** To clarify Q1 first: **Yes, the ground-truth set $\mathcal{A}$ consists of the final answer strings**, not the intermediate supporting facts.
>
> Regarding [W1], we respectfully provide three arguments supporting the validity and robustness of $R_{recall}$:
>
> * **Implicit Signal for Reasoning:** Using the final answer as the retrieval target is an effective proxy for multi-hop tasks. In multi-hop scenarios (e.g., HotpotQA), the final answer typically cannot be retrieved unless the agent has successfully identified and queried the intermediate "bridge" entities. Thus, the presence of the final answer implicitly signals that the necessary reasoning chain has been traversed.
> * **Discouraging Redundancy:** We specifically address the concern that $R_{recall}$ might fail to distinguish between precise and redundant queries. Empirically, we find that $R_{recall}$ implicitly penalizes redundancy. Since repeating a query yields identical documents (contributing zero marginal gain to recall), the agent learns to formulate distinct, diverse queries to maximize information coverage.
> * **Quantitative Evidence:** As shown in our granular behavioral analysis (Table below), DeSA reduces the Duplicate Query rate from 13.88% (Search-R1) to 5.30%, a significant 61.8% reduction. This demonstrates that the recall objective effectively drives the agent away from redundant loops and toward efficient exploration.
> * **Recall vs. Precision:** We also evaluated an Accuracy Reward ($R_{acc}$), which penalizes low-precision retrieval. As noted in our Ablation Study (Section 6.3.1), while $R_{acc}$ slightly improves simple QA, it degrades performance on complex Multi-hop QA. Multi-hop reasoning benefits more from broad information coverage (high recall) than from strict precision, which can prematurely constrain the agent's exploration scope.
>
> **Table: Comparison of Deficient Behaviors**
> | Deficiency Category | Search-R1 (Baseline) | DeSA (Ours) | Reduction Rate |
> | :--- | :---: | :---: | :---: |
> | Fail to Search | 0.78% | 0.11% | $\downarrow$ 85.9% |
> | Duplicate Queries | 13.88% | 5.30% | $\downarrow$ 61.8% |
> | Invalid Search | 20.34% | 2.09% | $\downarrow$ 89.7% |
> | **Total Deficient Rate** | **23.36%** | **6.96%** | **$\downarrow$ 70.2%** |
>
> ---
>
> > **[W2]:** The claim “simple recall works better” needs more support beyond final EM; need search behavior quality analysis.
>
> **[A2]:** We appreciate the reviewer's rigor. To substantiate our conclusion that "simple recall works better," we went beyond the final Exact Match (EM) scores and conducted an additional granular analysis of search behaviors on the full evaluation set ($N=51,712$). We compared our simple Recall reward ($R_{recall}$) against two alternative Stage-1 reward designs:
> * **Accuracy Reward ($R_{acc}$):** Rewards the precision of retrieval (proportion of retrieved docs containing the answer).
> * **Recall + Penalty:** Adds an explicit negative penalty for deficient behaviors to $R_{recall}$.
>
> | Method | Recall (%) | Fail to Search | Duplicate Queries | Invalid Searches | Total Deficient Rate |
> | :--- | :---: | :---: | :---: | :---: | :---: |
> | DeSA ($R_{recall}$) | 64.51% | 0.11% | 5.30% | **2.09%** | **6.96%** |
> | Ablation: $R_{acc}$ | **64.94%** | **0.01%** | 7.82% | 3.05% | 9.75% |
> | Ablation: $R_{recall}$ + Penalty | 62.91% | 0.26% | **4.88%** | 7.22% | 9.93% |
>
> **Analysis:**
> The simple $R_{recall}$ objective strikes the best balance. It maintains high recall while achieving the lowest Total Deficient Rate (6.96%), confirming that a dense, positive signal is most effective for the Search Skill Acquisition stage.
>
> ---
>
> > **[Q2]:** The choice to switch stages at “peak EM before collapse” seems post-hoc. Is there a principled way?
>
> **[A3]:** We acknowledge that monitoring the EM curve feels empirical, but it relies on a principled observation of the "objective shift" phenomenon. In Stage 1, the agent is optimizing a sub-objective (retrieval). The "collapse" point represents the moment where the model begins to over-optimize this sub-objective at the expense of its internal language modeling and reasoning coherence. Therefore, detecting the peak of the primary metric (EM) is a standard and principled **Early Stopping** strategy used in supervised learning and RL to prevent overfitting to a proxy or auxiliary task. We have included this rationale in the revised paper.

---

### Official Review · Reviewer_Qmee · 2025-10-28

**Soundness:** 3
**Presentation:** 3
**Contribution:** 3
**Rating:** 4
**Confidence:** 5

**Summary:**

The paper presents an interesting idea that identifies and categorizes different types of failed search behaviors in Search-R1, providing useful insights for developing more robust research agents.
The proposed method to mitigate these failures is conceptually valuable, and the writing is generally clear and easy to follow. In particular, the analysis of Search-R1’s failure cases is insightful and helps motivate the proposed approach.

However, the paper also suffers from several notable weaknesses. First, the recall reward design appears hacky and unstable, yet the paper lacks sufficient analysis or explanation of its behavior and training dynamics. Second, although the proposed method shows some improvement, the evaluation is superficial and misses a fine-grained breakdown across different failure categories, which limits the interpretability of the results. Third, the implementation details are incomplete—important training configurations and hyperparameters are not reported, making the work difficult to reproduce and evaluate fairly.

**Strengths:**

- The overall idea is interesting. The paper identifies and categorizes different types of failed search behaviors, which provides useful insights for building more robust research agents. The proposed method to mitigate these failures is conceptually valuable.

- The writing is generally clear and easy to follow. In particular, the analysis of Search-R1’s failure cases is very insightful and helps readers understand the motivation behind the proposed improvements.

**Weaknesses:**

- The recall reward design appears somewhat hacky. As shown in Figure 5, when the recall curve increases, the answer accuracy (EM) curve decreases. The authors manually select step = 50 based on observation, since continuing training leads to a sharp drop in EM. However, the paper provides no analysis of why the EM curve collapses after this point. What are the model’s outputs beyond that step? Would using a softer evaluation metric (e.g., F1 score or an LLM-as-a-judge metric) change this trend? It is also intriguing that when EM approaches zero, the recall curve remains high — suggesting that the model still generates good retrieval queries but fails to produce correct final answers. Some explanation or insight into this phenomenon would be valuable.

- Although it is encouraging to see that DeSA reduces the deficiency-search rate from 23.36% (Search-R1) to 6.96%, the analysis lacks finer-grained breakdowns. It would strengthen the paper to include both quantitative and qualitative analyses across different failure categories (e.g., Failed to Search, DQ, and IS), ideally through a case-by-case comparison with the original Search-R1 failure examples.

- The implementation details are insufficient. Important factors such as total training steps, learning rate, and other hyperparameters of the proposed model are not clearly reported. These should be included for reproducibility.

**Questions:**

Q1: Analysis of the Recall–Answer Trade-off:

The recall reward appears to cause instability in training — as recall improves, EM drops sharply (Figure 5).
1. Could the authors analyze why the EM curve crashes after step = 50?
2. What do the generated answers look like beyond that step?
3. Would the same trend persist if the evaluation used a softer metric (e.g., F1 score or an LLM-as-a-judge metric)?

Q2: Granular Evaluation of Failure Categories:

The reduction of the deficiency-search rate from 23.36% to 6.96% is promising, but more fine-grained analysis is needed.
1. Can the authors provide per-category breakdowns (Failed to Search, DQ, IS) and quantify improvements for each?
2. A case-by-case comparison with Search-R1’s failure examples would help verify whether the improvements are systematic or limited to specific types of errors.


Q3: Implementation Details:

The paper omits crucial hyperparameters (e.g., learning rate, total training steps, batch size). These should be provided, ideally in an appendix, to ensure reproducibility. The authors could also report the number of training examples and their distribution across different query types.

---

> ### Author Response · Authors · 2025-11-26
>
> We sincerely thank the reviewer for the constructive feedback and detailed questions. We appreciate the opportunity to clarify the training dynamics of our Stage 1 reward and to provide a more granular analysis of our results. We have addressed your questions item-by-item below.
>
> ---
>
> > **[Q1 & W1]:** Analysis of the recall–answer behaviors in Stage-1 (Why does EM crash? What do outputs look like?)
>
> **[A1]:** We thank the reviewer for this insightful observation and would like to elaborate further. The phenomenon shown in Figure 5 is a classic case of **Reward Hacking**.
>
> * **Why EM crashes:** In Stage 1, the agent is optimized solely based on retrieval recall. As training progresses beyond the "peak" point (approx. Step 50), the agent discovers that it can further maximize its reward by dedicating all its action budget ($n=4$ in our experiments) to search actions to ensure every possible relevant document is retrieved.
> * **Model Outputs:** Consequently, the model ceases to generate the final answer structure (e.g., the `<answer>` tags) entirely, viewing the answering phase as a "waste" of actions that does not contribute to the retrieval reward. This behavior causes the Exact Match (EM) score to decrease to near zero.
> * **Softer Metrics:** The collapsing trend will persist regardless of the specific evaluation metric used (e.g., F1 score or LLM-as-a-judge). As long as the reward metric is defined only by the coverage extent of search content, the model will inevitably learn to hack the objective by maximizing retrieval actions while abandoning answering.
>
> ---
>
> > **[Q2 & W2]:** Granular Evaluation of Failure Categories (Breakdowns and Case Studies).
>
> **[A2]:** We agree that a fine-grained analysis strengthens the paper. We have included a detailed breakdown of failure modes in the revised manuscript to demonstrate the systematic improvements achieved by DeSA.
>
> **Quantitative Analysis:**
> The table below compares the deficient search behavior rates of the baseline (Search-R1) versus DeSA (Qwen2.5-3B-Instruct) across the total evaluation set ($N=51,712$). DeSA significantly suppresses all types of deficient behaviors. Notably, Invalid Searches are reduced by nearly 90%, and Duplicate Queries are reduced by over 60%.
>
> | Deficiency Category | Search-R1 (Baseline) | DeSA (Ours) | Reduction Rate |
> | :--- | :---: | :---: | :---: |
> | Fail to Search | 0.78% | 0.11% | $\downarrow$ 85.9% |
> | Duplicate Queries | 13.88% | 5.30% | $\downarrow$ 61.8% |
> | Invalid Search | 20.34% | 2.09% | $\downarrow$ 89.7% |
> | **Total Deficient Rate** | **23.36%** | **6.96%** | **$\downarrow$ 70.2%** |
>
> *(Note: Percentages represent the proportion of trajectories exhibiting the specific failure mode relative to the full dataset. The column sum exceeds the Total Deficient Rate because a single trajectory may contain multiple types of errors, such as both duplicate and invalid queries.)*
>
> **Qualitative Case Study:**
> We have added a comprehensive Qualitative Analysis section (**Appendix C**) to the revised paper, providing side-by-side trajectory comparisons. In Appendix C.1, we illustrate how DeSA overcomes specific baseline failures.
>
> ---
>
> > **[Q3 & W3]:** Implementation Details (Hyperparameters).
>
> **[A3]:** We respectfully clarify that the key hyperparameters (including learning rate, batch size, and total training steps) were documented in **Appendix A** of the original submission. Furthermore, following the reviewer’s valuable suggestion to improve data transparency, we have added the detailed statistics of training examples (including the number of entries and their distribution across different query types) in the newly added **Appendix D**.

---

### Official Review · Reviewer_oNix · 2025-10-31

**Soundness:** 2
**Presentation:** 2
**Contribution:** 3
**Rating:** 4
**Confidence:** 5

**Summary:**

This paper addresses the limitations of training search-augmented LLM agents solely with outcome-based rewards (e.g., exact match). The authors propose DeSA, a two-stage training framework that first trains agents on search effectiveness using recall-based rewards, then optimizes answer generation with outcome rewards. Experiments across seven QA benchmarks show that DeSA reduces deficient search rates and improves both search recall and final-answer accuracy compared to single-stage outcome-only training.

**Strengths:**

* **Intuitive solution**. The two-stage decoupling approach is conceptually straightforward and well-motivated by the sequential nature of search-then-answer tasks.
* **Improved results**. Testing across seven diverse QA benchmarks provides the evidence for the method's effectiveness.

**Weaknesses:**

1. **Weak motivation-solution alignment**. The motivation analysis discusses issues listed in Figure 1, including Fail to Search (Skip Search), Duplicate Queries, and w/ Invalid Searches. However, Skip Search is due to knowledge being within the parameters or hallucination; Duplicate Queries may be due to reward hacking and entropy collapse; and w/ Invalid Searches is due to the agent generating an invalid search query format. These issues can be addressed by applying a retrieval reward, a repeat penalty, and a format reward. The paper claims that improving recall indirectly enhances search capabilities, yet it fails to establish clear causal links between the identified problems and the proposed remedy.
2. **Recall robustness**. In Eq. 4, the authors use string matching to calculate the recall reward. This method does not address practical challenges such as spelling variations, abbreviations, etc. These issues could significantly degrade the quality of the reward signal.
3. **Insufficient analysis of single-stage alternatives**. Comparison with single-stage training in Section 6.3.2 using combined rewards may be unfair due to suboptimal hyperparameter choices. Different weighting ratios or more sophisticated reward combination strategies should be explored before concluding that decoupling is necessary.
4. **Missing error analysis and failure modes**. The paper lacks detailed post-training error distributions and concrete failure case studies. Without understanding when and why DeSA fails, it's difficult to assess the method's limitations and potential improvements.
5. **Incomplete latest backbones**. The absence of comparisons with Qwen3 models  is notable, especially given the rapid progress in base model capabilities that might naturally address some search deficiencies.
6. **References**. Multiple citation formatting errors throughout the paper (missing conference venues and arXiv identifiers).

**Questions:**

Line 372 & 376: Where are Figures 3a and 3b referenced in the text?

---

> ### Author Response · Authors · 2025-11-26
>
> We sincerely thank the reviewer for the detailed feedback and valuable suggestions. We appreciate the opportunity to clarify our motivation and strengthen our experimental analysis. We have conducted the suggested experiments and analysis. Below, we provide itemized responses to the concerns raised.
>
> ---
>
> > **[W1]:** Weak motivation-solution alignment.
>
> **[A1]:** We appreciate the reviewer's thoughtful breakdown of the deficient search behaviors. We realize that our initial presentation may have obscured the full scope of DeSA's motivation, which is twofold: (1) to mitigate deficient search behaviors caused by outcome-reward only training, and (2) to explicitly acquire effective search skills before answer optimization.
>
> To directly answer the reviewer's question regarding whether specific constraints (e.g., penalties) suffice, we conducted an additional experiment using Outcome Reward (EM) + Behavior Penalties (penalizing duplicate queries, failing to search, and invalid search with -0.2 per each occurrence). We used Qwen2.5-3B-Instruct as the backbone model and kept the training and evaluation config the same as DeSA. The results are compared with DeSA in the table below:
>
> | Method | NQ | TriviaQA | PopQA | HotpotQA | 2Wiki | Musique | Bamboogle | Avg. |
> | :--- | :---: | :---: | :---: | :---: | :---: | :---: | :---: | :---: |
> | EM + Penalties | **0.387** | **0.578** | **0.404** | 0.345 | 0.356 | 0.121 | 0.266 | 0.351 |
> | **DeSA (Ours)** | 0.375 | 0.575 | 0.397 | **0.352** | **0.363** | **0.134** | **0.347** | **0.363** |
>
> **Analysis & Conclusion:**
> As shown in the table, the "EM + Penalties" performs slightly better than DeSA on General QA datasets (e.g., NQ, TriviaQA, PopQA). This indicates that for simpler, single-hop retrieval tasks, negative constraints with EM reward are sufficient to regularize and guide the model.
>
> However, **a significant performance gap emerges in complex Multi-Hop QA tasks**. DeSA consistently outperforms the penalty-based baseline on HotpotQA, 2WikiMultiHopQA, Musique, and Bamboogle. Notably, on Bamboogle, the penalty-based agent lags behind DeSA by a substantial margin (-8.1%). This validates our motivation: while penalties successfully suppress specific "bad" behaviors (symptoms), they fail to teach the agent how to conduct effective multi-step exploration (root cause). In complex scenarios, the sparse outcome signal (even with penalties) is insufficient to guide the agent through the reasoning chain. DeSA’s explicit search training (Stage 1) is therefore essential for acquiring the robust search capabilities required for multi-hop reasoning.
>
> ---
>
> > **[W2]:** Recall Robustness
>
> **[A2]:** We thank the reviewer for this insightful suggestion. While Eq. 4 is based on string matching, we would like to clarify that our evaluation relies on standard datasets (e.g., Natural Questions) that provide lists of acceptable answer variants (aliases, abbreviations, etc.). Furthermore, we apply standard normalization (e.g., removing punctuation and articles) to all ground truths to handle formatting differences. Our empirical observations suggest that the current Recall metric serves as a favorable trade-off between effectiveness and computational efficiency. We acknowledge that incorporating finer-grained metrics like F1 score or LLM-based evaluation is a valuable direction and will leave it for future work.
>
> ---
>
> > **[W3]:** Insufficient analysis of single-stage alternatives
>
> **[A3]:** We thank the reviewer for pointing out the need to explore different weighting ratios. To verify if the performance gap stems from suboptimal hyperparameters, we evaluated the single-stage baseline with varying recall weights $w \in \{0.2, 0.5, 0.8\}$ (see Table below).
>
> | Method | NQ | TriviaQA | PopQA | HotpotQA | 2Wiki | Musique | Bamboogle | **Avg.** |
> | :--- | :---: | :---: | :---: | :---: | :---: | :---: | :---: | :---: |
> | **DeSA (Ours)** | 0.375 | **0.575** | 0.397 | **0.352** | **0.363** | **0.134** | **0.347** | **0.363** |
> | Single-Stage ($w=0.2$) | 0.382 | 0.573 | **0.401** | 0.349 | 0.337 | 0.130 | 0.323 | 0.356 |
> | Single-Stage ($w=0.5$) | **0.386** | 0.563 | 0.383 | 0.348 | 0.330 | 0.132 | 0.307 | 0.350 |
> | Single-Stage ($w=0.8$) | 0.329 | 0.540 | 0.375 | 0.306 | 0.304 | 0.098 | 0.218 | 0.310 |
>
> The results show that even the best-performing single-stage variant ($w=0.2$) lags behind DeSA in average performance (35.6% vs. 36.3%). Crucially, DeSA maintains a clear lead on complex reasoning benchmarks (e.g., 2WikiMultiHop, Bamboogle), where the conflict between retrieval and generation is most pronounced. This confirms that decoupling provides a structural benefit that cannot be fully achieved by simply tuning reward weights in a single-stage framework.

---

> ### Author Response · Authors · 2025-11-26
>
> ---
>
> > **[W4]:** Error Analysis of DeSA
>
> **[A4]:** We thank the reviewer for the constructive suggestion. We have updated **Appendix C.2** with a detailed failure mode analysis, identifying two primary categories:
>
> * **Temporal Mismatch (Question-Knowledge Alignment):** Errors where the static nature of the retrieval corpus conflicts with the temporal intent of the question. For instance, when questions ask for the "next" event, the model may fail to ground the answer in the question’s context, leading to hallucinations.
> * **Reasoning & Comprehension Gaps:** Instances where the model retrieves the correct context but misinterprets the semantic constraints of the question. A typical error pattern is answering a temporal question (e.g., asking for a duration) with a location entity, indicating a lack of fine-grained reading comprehension.
>
> ---
>
> > **[W5]:** Incomplete latest backbones.
>
> **[A5]:** We appreciate the reviewer’s suggestion regarding the rapid evolution of base models. To verify if stronger backbones naturally mitigate the need for DeSA, we implemented both our method (DeSA) and the baseline (Search-R1) using the rl-factory framework, utilizing Qwen3-4B as the backbone. We have completed the initial validation for these models. As shown in the table below, DeSA consistently outperforms the strong baseline across most datasets.
>
> | Dataset | Search-R1 (Mean@1) | DeSA (Mean@1) |
> | :--- | :--- | :--- |
> | NQ | 0.3886 | **0.3934** |
> | TriviaQA | **0.6028** | 0.5915 |
> | PopQA | 0.4267 | **0.4387** |
> | HotpotQA | 0.3747 | **0.3799** |
> | 2WikiMultihopQA | **0.3898** | 0.3757 |
> | Musique | 0.1612 | **0.1712** |
> | Bamboogle | 0.4160 | **0.4480** |
> | **Average** | 0.3943 | **0.3998** |
>
> These results suggest that although stronger backbones naturally exhibit less deficient search behavior, **explicit recall training remains highly effective in further enhancing performance.**
>
> ---
>
> > **[W6 & Q]:** Reference formatting and Figure numbering (Line 372 & 376).
>
> **[A6]:** Thanks to the reviewer for the suggestion. We have corrected all citation formats and the figure number issue in our revised version.

---

### Official Review · Reviewer_WWWA · 2025-11-03

**Soundness:** 2
**Presentation:** 3
**Contribution:** 2
**Rating:** 2
**Confidence:** 5

**Summary:**

The paper introduces an extension to the Search-R1 framework by decomposing the end-to-end pipeline to a two-stage pipeline, one for search and one for answer generation. The reward for search is defined by recall of answer being presented in the retrieved documents.

**Strengths:**

+ The paper focuses on a timely topic and Search-R1 seems like a reasonable framework for exploring the idea.
+ The paper is easy to follow.
+ Extensive experiments are conducted on seven datasets.

**Weaknesses:**

- I do like the general idea of having reward models that evaluate the retrieval part of the Search-R1. However, I don't think this paper does a good execution of this idea. First, retrieval happens multiple times in Search-R1 and reward seems to be outcome-based reward. So if the model repeats the same query over and over, there is no penalty for the model. I see that the results look at this and shows the model reduces duplication however duplication is just the extreme case. Also I'd like to see more modeling contribution on that front or at least theoretical justification on why outcome-based reward would lead to this observation.

- Another thing that I don't like about the execution of the general idea in this work is that end-to-end training enables us to do much better in complex tasks. I am surprised that the authors decided to tear apart the end-to-end training pipeline of Search-R1. I think this is a poor design choice and I encourage them to focus on better reward modeling for end-to-end training. Note that I do not base my overall recommendation based on this comment, because one may disagree with my comment and this is mostly a suggestion for future work.

- I am surprised that the authors decided to run their experiments on Qwen instruct models. Multiple RL papers, including the Search-R1 paper which is the basis for this work, suggest that base models are more suitable for RL training. The Search-R1 results with Qwen 7B base is 0.431 on all seven datasets, about 4% higher than the highest results presented by DeSA in this work.

- Statistical significance tests are required to demonstrate meaningful improvements compared to baselines (at least compared to Search-R1).

- Minor: why Search-o1 result is bold-faced in Table 2? Maybe a mistake...

**Questions:**

See the weaknesses.

---

> ### Author Response · Authors · 2025-11-26
>
> We sincerely thank the reviewer for the constructive suggestions, which have significantly improved our paper. We have completed all suggested experiments and updated our manuscript. Below we provide itemized responses to address the reviewer’s concerns.
>
> ---
>
> > **[Q1]:** Duplication is just the extreme case. I'd like to see more modeling contribution on that front or at least theoretical justification on why outcome-based reward would lead to this observation.
>
> **[A1]:** As shown in Figure 3, the proportion of trajectories exhibiting duplication on the 3B and 7B models is $20.1\%$ and $7.0\%$, respectively, which are considerably high percentages in the cases where the agent failed to recall the answer. Also, duplicate querying is only one of several deficient search behaviors we identified. Therefore, we conclude that such deficient behaviors are not extreme cases but a frequently observed pattern in outcome-reward trained search agents. In our revised version, we also visualized the impact of deficient search behaviors on agent performance in Figure 2, which causes significantly lower Recall and EM rate.
>
> **Theoretical Justification:**
> The core issue is the **sparsity of the outcome reward** relative to the vast exploration space of search actions. The probability of generating a perfectly matched answer and receiving a positive signal ($R=1$) is low for complex queries (like multi-hop QA). In the context of GRPO with a limited group size (e.g., $G=5$ or similar), this means that for many difficult samples, the entire group may fail to retrieve the correct answer, resulting in uniform zero rewards. As a result, the relative advantage becomes uninformative, leading to training instability and inefficient exploration.
>
> Besides, if lacking a clear gradient signal to distinguish "better retrieval" from "poor retrieval," the policy struggles to escape its initialization priors. Consequently, it is prone to drifting toward low-entropy behaviors inherent to the base model, such as repetition (Duplicate Queries) or inaction (Fail to Search). Stage 1 of DeSA resolves this by introducing a **dense reward** ($R_{recall}$). This ensures that good search attempts receive a positive gradient, effectively "activating" the agent’s search capabilities and guiding it out of these sub-optimal behavioral loops.
>
> ---
>
> > **[Q2]:** Tear apart the end-to-end training pipeline of Search-R1.
>
> **[A2]:** We appreciate the reviewer's opinion on end-to-end training. We believe DeSA offers critical insights into the RL agents’ training pipeline, rather than simply tearing apart the end-to-end paradigm. Our experiments confirm that for complex agentic tasks (characterized by long trajectories and multiple interaction turns), relying solely on outcome rewards exacerbates the credit assignment problem, leading to the deficient search behaviors we observed.
>
> By introducing a specific search reward, we successfully teach the model how to search before asking it to answer. We believe that designing such dense rewards is an effective way to address the low exploration efficiency of RL agents on difficult tasks. By enabling the agent to better learn and master sub-tasks, this approach has the potential to achieve more efficient and truly effective RL exploration.
>
> ---
>
> > **[Q3]:** Choice of backbone model: Instruct Model vs. Base Model.
>
> **[A3]:** We sincerely thank the reviewer for this insightful suggestion. We agree that using base models as backbones is crucial for validating the effectiveness of DeSA. Following your advice, we conducted additional experiments using **Qwen2.5-3B-Base** as the backbone.
>
> The results, presented in the table below, show that DeSA achieves **even more significant improvements on the base model** compared to the baseline (Search-R1). Specifically, DeSA raises the average performance from **0.303** to **0.396**, a relative improvement of **30.7%**. Notably, the 3B DeSA’s performance even comes close to Search-R1 with 7B backbone, demonstrating that our decoupled training framework is highly effective at instilling search capabilities into base models from scratch.
>
> | Method | NQ | TriviaQA | PopQA | HotpotQA | 2Wiki | Musique | Bamboogle | **Avg.** |
> | :--- | :---: | :---: | :---: | :---: | :---: | :---: | :---: | :---: |
> | Search-R1 (Base) | 0.406 | 0.587 | 0.435 | 0.284 | 0.273 | 0.049 | 0.088 | 0.303 |
> | **DeSA (Base)** | **0.470** | **0.625** | **0.448** | **0.396** | **0.365** | **0.145** | **0.323** | **0.396** |

---

> ### Author Response · Authors · 2025-11-26
>
> ---
>
> > **[Q4]:** Statistical significance tests
>
> **[A4]:** We appreciate the reviewer’s suggestion to verify the statistical reliability of our results. To rigorously evaluate the improvements of DeSA over the Search-R1 baseline, we conducted Micro-Average Z-tests using the exact evaluation set sizes used in our experiments ($N=51,713$ samples in total). The results confirm that DeSA provides highly statistically significant improvements across all model settings ($p < 10^{-9}$).
>
> Specifically, the Qwen2.5-Base model demonstrated the most substantial gain ($Z=19.52, p < 10^{-84}$). The Instruct models also showed robust performance: the 7B-Instruct and 3B-Instruct models achieved Z-statistics of $7.11$ ($p < 10^{-12}$) and $6.23$ ($p < 10^{-9}$) respectively. These results indicate that the aggregate performance improvement provided by DeSA is statistically robust and practically meaningful across the full distribution of queries.
>
> ---
>
> > **[Q5]:** Minor issue in Table 2
>
> **[A5]:** We thank the reviewer for careful attention to detail. We have corrected this mistake in the revised version of the manuscript to ensure that only the best-performing results are highlighted.

---

### Author Response · Authors · 2025-12-02

Dear ACs, SACs, PCs, and Reviewers:

We sincerely thank the reviewers for their constructive feedback. We are encouraged that the reviewers recognized **the intuitiveness and simplicity of our decoupled framework** (Reviewer oNix, 8b1L), **the insightful categorization of deficient search behaviors** (Reviewer Qmee), and **the comprehensive experimental validation across diverse benchmarks** (Reviewer WWWA, 8b1L).

To further strengthen our contributions and address specific concerns, we have updated the manuscript with extensive new experiments. Key highlights include:

1. **Significant Gains on Base Models:** DeSA achieves a **30.7% relative improvement** on Qwen2.5-3B-Base compared to the Search-R1 baseline, proving our method effectively instills search capabilities from scratch without relying on instruction tuning.
2. **Generalization to Strong Backbones:** New experiments on **Qwen3-4B** confirm that DeSA consistently outperforms baselines even with stronger foundation models, validating that explicit recall training remains essential.
3. **Validation of Decoupled Design:** We empirically demonstrated that our **two-stage decoupling** significantly outperforms single-stage baselines with complex reward shaping (e.g., (1-w) $R_{outcome} + w R_{recall}$), particularly on multi-hop reasoning tasks.
4. **Robustness:** DeSA reduces deficient search behaviors (e.g., duplicate/invalid queries) by **~70%** (from 23.36% to 6.96%). All improvements are statistically significant ($p < 10^{-9}$).

**Conclusion:** Beyond specific performance gains, our work highlights a critical insight for the broader agentic research community: **decoupling complex agentic workflows into sub-task training stages (e.g., Search then Answer) is a highly scalable and effective strategy.** This paradigm effectively mitigates the credit assignment problem inherent in long-horizon RL, offering a generalizable roadmap for training future agents on increasingly complex tasks.

Best regards,
The Authors

---

### Meta-Review · Area_Chair_JDos · 2026-01-07

**Summary:**

This paper proposes DeSA, a two-stage RL training scheme for search-augmented QA agents: first train search with a recall-style reward, then train answering with outcome reward (EM). Reviewers agreed the decoupled idea is intuitive and the failure categorization for Search-R1-style agents is useful, but questioned (i) whether the recall reward is too coarse / “hacky”, (ii) whether decoupling is truly necessary versus better single-stage reward design, and (iii) whether results are reliable given backbone choices and missing significance/implementation details. The rebuttal adds several important experiments (base-model backbones, Qwen3-4B, single-stage weight sweeps, penalties baseline), provides a clearer analysis of reward hacking in stage 1, and includes statistical significance tests and finer-grained deficiency breakdowns.

**Reviewer Concerns:**

Addressed by rebuttal:
- Backbone choice (instruct vs base): Added base-model experiments showing sizable gains over Search-R1 on Qwen2.5-3B-Base, and additional tests on Qwen3-4B.
- Need for statistical evidence: Added significance testing and fixed at least one table-formatting issue.
- “Hacky” recall reward / EM collapse: Explained the stage-1 collapse as reward hacking (agent spends all budget on search), and justified early stopping + stage-2 outcome optimization.
- Decoupling vs single-stage: Added comparisons to single-stage combined rewards (with weight sweeps) and to EM+behavior penalties; decoupling remains stronger especially on multi-hop datasets.
- Granular behavior analysis: Added per-category deficiency rates (fail-to-search / duplicate queries / invalid searches) and reduction magnitudes, plus qualitative examples in an appendix.

Still outstanding / limitations:
- Recall reward coarseness for multi-hop: The recall reward is still based on final answer strings (not intermediate facts), which remains a legitimate limitation; the rebuttal argues it is a useful proxy but it’s not fully satisfying for all multi-hop settings.
- Stage transition heuristic: Switching at “peak EM before collapse” is defended as early stopping, but still feels a bit ad hoc; an automatic criterion would strengthen the story.
- Net gains on stronger backbones are modest: On Qwen3-4B, improvements are small and mixed across datasets; the benefit is present but not uniformly strong.

**Reviewer Scores:**

Estimated score updates after discussion:
- WWWA: 2 → 3–4 (base-model + significance + design justification likely addresses core objections, though they may still prefer end-to-end reward modeling).
- oNix: 4 → 4–5 (many requested baselines and analyses were added).
- Qmee: 4 → 5 (reward-hacking explanation + granular breakdown + hyperparams address main points).
- 8b1L: 4 → 4–5 (answered reward definition and provided broader behavior metrics; still cautious about recall reward design).

---

### Decision · Program_Chairs · 2026-01-26

Reject